# A High Cross-Individual Accuracy EEG-based Seizure Detection Algorithm Based on Multiple Source Domain Adaption

Haiting Li

*School of Information and Communication Engineering*
*University of Electronic Science and Technology of China,UESTC*
Chengdu, China
202211012427@std.uestc.edu.cn

Xinyang Deng

*Glasgow College*
*University of Electronic Science and Technology of China,UESTC*
Chengdu, China
2021190908009@std.uestc.edu.cn

Yushan Li

*School of Information and Communication Engineering*
*University of Electronic Science and Technology of China,UESTC*
Chengdu, China
202221011622@std.uestc.edu.cn

Jun Zhou

*School of Information and Communication Engineering*
*University of Electronic Science and Technology of China,UESTC*
Chengdu, China
zhouj@uestc.edu.cn

*Abstract*—**Electroencephalography (EEG) has been proven to be very effective in seizure detection. However, individual variability has severely limited its practical use. Due to differences in brain structure and skin, EEG signals can vary greatly from one individual to another, which leads to a low cross-individual seizure detection accuracy. To solve the problem, two methods are proposed in this work. Firstly, by treating cross-individual tasks as transfer learning scenarios, template matching based on multiple source domain adaption neural network method is proposed. The method selectively eliminate differences between multiple source domains and target domain to improve the accuracy. Secondly, considering the fact that seizure data is much less than non-seizure data, adaptive calibration data select based on average Pearson correlation coefficient with principal component analysis method is proposed. The source domain seizure data is adaptive selected to alternate the calibration seizure data. With the method, the accuracy further improved. The proposed methods are validated on CHB-MIT EEG data set to achieve state-of-the-art performance with 85.21% sensitivity, 93.76% specificity, and 92.50% accuracy.**

*Keywords—EEG, seizure detection, cross-individual, template matching, domain adaption*

## I. INTRODUCTION

Epilepsy, a neurological disorder characterized by recurrent seizures, affects millions of individuals worldwide, making accurate and timely diagnosis essential for treatment. Among the various diagnostic tools available, electroencephalography (EEG) stands out due to its non-invasive nature and its ability to provide real-time monitoring of brain activity. EEG works by recording the electrical activity of the brain through electrodes placed on the scalp, offering a direct insight into the brain's functioning. This makes EEG particularly valuable for detecting abnormal brain wave patterns associated with seizure.

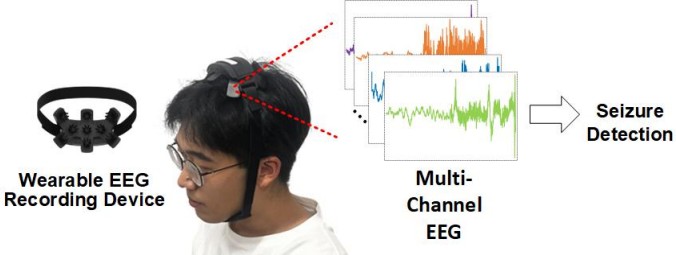

Fig. 1. Wearable EEG recording devices to capture EEG for seizure detection.

Compared to other diagnostic methods such as magnetic resonance imaging (MRI) or computed tomography (CT) scans[1][2], EEG offers several distinct advantages in seizure detection. While MRI and CT scans provide detailed images of brain structure, they do not capture the dynamic electrical activity that characterizes seizures. EEG, on the other hand, can detect even brief and subtle changes in brain activity, making it highly sensitive to the electrical disturbances that occur during seizure events. Additionally, EEG can be used for prolonged monitoring, which is crucial for capturing sporadic seizure activity that might be missed in a short-term scan[3]. Finally, the final advantage of using EEG for seizure detection is the ability to perform real-time detection on wearable devices, which improves more application scenarios, as shown in Fig.1.

Despite these benefits, a significant challenge in utilizing EEG for seizure detection is the considerable individual variability in EEG patterns. Each person's brain wave patterns can be greatly different from each other due to factors such as genetics, age, and the presence of other neurological conditions. This high degree of subject-specific variability restricts the development of universal algorithms for accurate seizure detection. For instance, an EEG pattern considered normal for

one individual can indicate seizure activity in another, leading to potential misdiagnoses.

Previous research has primarily concentrated on individual-specific seizure detection, which necessitates the collection and feature extraction of data for each individual individual [4][5][6]. Although these algorithms often achieve high accuracy, the process of gathering extensive EEG signals and manually labeling them will cost time and manpower. Furthermore, It is still difficult to difficult to complete EEG signal generation because of the unpredictable variability[7][8]. The internal mechanism of this difference has not been studied clearly yet. Consequently, creating algorithms for cross-individual seizure detection continues to be a major research challenge.

Many work also considered online calibration for cross-individual seizure detection[9][10]. With the acceptable small amount of calibration data to complete model update, those work received a considerable increase of detection accuracy compared with no calibration data. However, the work only considered to use both seizure data and non-seizure data for online calibration. In real-world seizure detection, non-seizure data is much more easily available than seizure data. It may take a long time for a subject's first seizure to arrive, and the data for non-seizure may already abundant before the seizure arrives. At the same time, if the calibration is not completed until the first seizure, the accuracy of the first detection of seizure will not be guaranteed. Therefore, the online calibration should be done before the first seizure with only non-seizure data, which will make it more feasible in practical use.

In order to solve the problem mentioned above. In this work, firstly, by treating cross-individual seizure detection task as transfer learning scenario, template matching based on multiple source domain adaption neural network is proposed for selectively eliminate differences between multiple source domains and target domain to improve the accuracy. Secondly, considering the fact that seizure data is much less than non-seizure data, adaptive calibration data select based on average Pearson correlation coefficient of with principal component analysis method is proposed. With the method, the problem of no seizure calibration data is well solved and the detection accuracy is further improved.

The rest of the paper is organized as follows. In Section II, related work about cross-individual seizure detection are reviewed. In Section III, the two proposed methods are introduced in detail. In Section IV, the experimental setup, data set description and experimental results are presented. And the conclusion is in Section V.

## II. Related Work

Over the past decade, numerous machine learning techniques have been developed to enhance cross-individual seizure detection, often evaluated using leave-one-out cross-validation. For instance, Chen et al. used DWT to construct feature vectors from EEG segments and employed SVM as the classifier [11]. Testing on 18 individuals from the CHB-MIT dataset, they achieved a peak average accuracy of 92.30% with seven features on the coif3 wavelet. Jiang et al. applied k-NN, reducing dimension of individual features to harmonize training and test sets, resulting in a 74.03% average accuracy [12]. Similarly, Fergus et al. achieved 88% sensitivity and specificity using a k-NN approach without prior individual knowledge, with an 80/20 train-test data split [9].

In contrast to the work applied machine learning mentioned above, which require significant manual feature extraction method, deep learning techniques with end to end deep neural networks avoid the problem of feature extraction. A robust recurrent convolutional neural network is developed in [13], achieving 85% average sensitivity. Hossain et al. employed a deep CNN to extract spectral and temporal EEG features, surpassing state-of-the-art accuracy in cross-individual detection and achieving 99.65% in individual-specific detection [14]. Zhou et al. proposed a self-organizing fuzzy logic (SOF) classifier, which, through varied distance and granularity experiments, reached 93.02% sensitivity and 91.24% specificity in individual-specific tasks [15]. However, in cross-individual detection, dividing 24 individuals into six groups and using leave-one-group-out validation, the method achieved only 84.67% sensitivity and 82.06% specificity. Nasiri et al. utilized GANs to learn transferable features across individuals, improving sensitivity and specificity by 4%-5% and reducing latency [16]. In [10], an adaptive fine-grained template matching is used to adapt template in the process of test, and achieve 84.75% sensitivity and 92.31% specificity in cross-individual task.

## III. Methods

### A. Template Matching based on Multiple Source Domain Adaption Neural Network (TM-MDAN)

As mentioned in Section.I, the EEG signal is highly affected by individual variability, which limited cross-individual seizure detection accuracy. A model with a high individual-specific accuracy may be difficult to have a high cross-individual accuracy, so a calibration method for the model needs to be introduced. To address this issue, a template matching based on multiple source domain adaption neural network method (TM-MDAN) is proposed.

To begin with, the cross-individual seizure detection task is considered as a transfer learning scenario in this paper. In this task, subjects from train set is considered as source domain, which is the knowledge already acquired. In cross-individual seizure detection task, the source domain generally involves multiple individuals, and it will be denoted as $S = \{x^s, y^s\}$, where $x^s$ represents subjects in the source domain, and $y^s$ is the corresponding label. Subject from test set is considered as target domain, which is a specific subject to test. In our method, target domain is considered to include two parts because of the online calibration. The first part is the labeled calibration data which will be denoted as $Tl = \{x^{tl}, y^{tl}\}$. The amount of

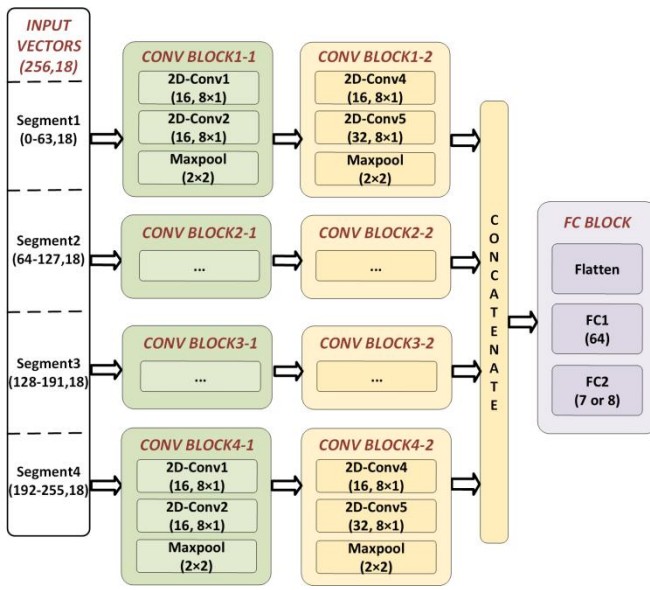

Fig. 2. The structure of our proposed baseline neural network with input vector segmentation.

samples in $Tl$ is small, and in our work, it is set at 0.5% of the whole test set, which is consistent with [1]. The second part is the unlabeled data in the test set that denoted as $Tu = \{x^{tu}\}$, which is 99.5% of the test set.

In this paper, a convolution neural network (CNN) is introduced as a baseline model. The structure of the CNN is shown in Fig.2. In the structure, Input vector segmentation is used, which has been proved in [17] to effectively reduce the overall model complexity without decreasing accuracy. The segmentation method ignores part of marginal effects that do not reduce accuracy, lead to a effectively reduced in model complexity. Specifically, after window segment, each window has 256 samples, and it will be segmented to 4 blocks with 64 samples each. For each input block, it will be put into the same four convolution layers and two max pooling layer. After convolution layers, the feature map vectors will be concatenated into one vector to input the flatten layer and two full-connected layers. It should be mentioned that FC1 in this structure will be treated as template layer which will be introduced in the rest part of section III.A. Each layer in the structure generate with a batch-normalization layer and 0.05 dropout layer.

Unlike other methods which use the vector output from the last layer of neural network as a probability to complete the classification, In this work, a template matching module is proposed after the neural network. The template matching will use calibration data to complete model update and as the basis for the final classification while the neural network is mainly used for feature extraction. The details are introduced below. During the calibration, the neural network can be considered as a feature extraction function $f$. The input of the function is the input of the neural network, the output of this function is the output feature map of a specific layer which will be considered as template layer. The output of this function, that is, the input of the template matching module, is $M = f(x)$. For a specific calibration data $x_c^{tl} \in Tl$ where $c$ is the class, we have $M_c^{tl} = f(x_c^{tl})$.

The data dimension of the feature vectors $M_c^{tl}$ is the same as the number of neurons in the template layer of the network. The template of each class $Tem_C$ is obtained and stored for later testing by calculating the mean value of all the feature vectors $M_c^{tl}$, that is expressed as below (1):

$$Tem_C = \frac{\sum_{i=1}^{N_c} M_{c,i}^{tl}}{N_c} \qquad \#(1)$$

Where $N_c$ is the number of samples of the class $c$ in $Tl$. During testing, the test data $x^{tu}$ are fed into the neural network and feature vectors are generated as $M^{tu} = f(x^{tu})$. The Euclidean distance $d_c$ between $M^{tu}$ and template of each $Tem_c$ is then calculated. After that, the minimum Euclidean distance $d_{ck}$ for all the classes is obtained, which completes the classification. (i.e. $c_k$ is the final classification result.).

Template matching uses the neural network as feature extraction, and since neural network is only trained by source domain data, the feature extraction capability are source domain related. With the features extracted, seizure and non-seizure can be well distinguished in the source domain. However, if the features itself has individual variability, it may not have a good capability to distinguish between seizure and non-seizure in the target domain. Therefore, if the network can extract features that are invariant with individuals, it can be better used for the target domain. Based on the analysis above, a multiple source domain adaption neural network is proposed for cross-individual seizure detection.

Domain adaption techniques in transfer learning are used to reduce domain shift by aligning feature distributions between the source and target domains. By employing adaption training in the neural network, the method help the model to learn domain-invariant features, enhancing its ability to generalize to new data. And the adaption training is applied in the multiple source domain adaption neural network proposed in this paper.

The multiple source domain adaption neural network used in this paper includes one feature extractor, one label classifier and multiple domain classifiers, as shown in Fig.3. The label classifier and the domain classifiers are connected to the feature extractor. The feature extractor is shared by the label classifier and domain classifiers. The branch of feature extractors combined with label classifier is the baseline CNN model introduced in Section III.A paragraph.3. This branch is for the classification of the class of seizure or non-seizure. However, the branch of feature extractors combined with each

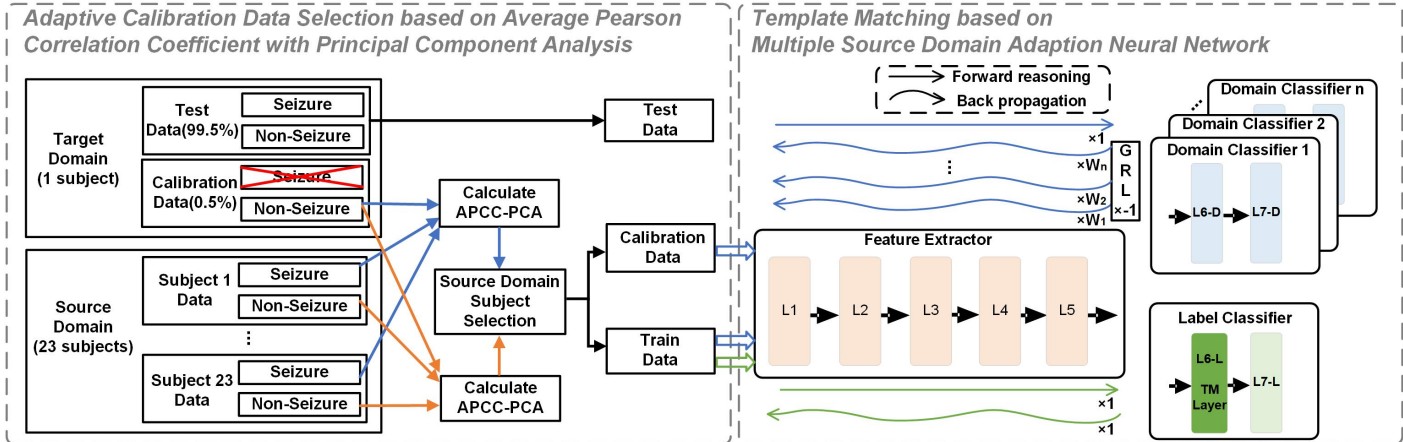

Fig. 3. The overall structure of our proposed method including Template Matching based on Multiple Source Domain Adaption Neural Network and Adaptive Calibration Data Selection based on Average Pearson Correlation Coefficient with Principal Component Analysis.

domain classifier is for domain adaption. These branches is for domain adaption for multiple source domains and one target domain. Each domain classifier has a gradient inverse layer (GRL) as input. The GRL does not perform any operation on the data in the forward calculation of the neural network. But in the process of back-propagation training, the GRL will reverse the gradient generated by domain classifier and then pass to feature extractor. GRL will make the training of feature extractor not in the direction of domain classifier loss function convergence, but in the direction of non-convergence. This will make the features extracted by feature extractor unable to help domain classifier identify the target domain or the source domain. Therefore, the training process above achieves the goal of training feature extractor to extract more individual invariant features.

In our scenario, the source domain consists of multiple individuals with differences that can be considered as multiple different source domains. Therefore, multiple domain classifier is applied and the number of domain classifiers is decided by the number of subject in the train set. The first advantage of multiple domain classifiers is to separate the internally different individuals of the source domain, and let them calculated gradient at the same time to avoid the overwriting of the training results of different individuals because of the sequence. Another advantage of multiple domain classifiers is that gradient generated from different source domains can be multiplied with different weight $W_i$. In that case, the influences of different source domain used for domain adaption can be controlled in different degree. Because the degree of correlation between individuals is different, and increasing the influence of individuals in source domain have more similar data to target domain can better improve the accuracy. The way of choose best similar subjects will be introduced in III.B.

During the calibration training process, in each data batch, the target domain calibration data $Tl$ is combined with different source domain data $S_i$ where i is the index of different source domain. As for label classifier branch, $S_i$ has its own seizure or non-seizure label will be used for label classifier training, and $Tl$ will not be used for training label classifier. As for domain classifier branches, each data in $S_i$ will be labeled '0' while each data in $Tl$ will be labeled '1' for training. Both gradient from label classifier and domain classifiers will be send to feature extractor at the same time.

With the proposed template matching based on multiple source domain adaption neural network method, the cross-individual detection accuracy increase greatly to show the advantages of our method.

## B. Adaptive Calibration Data Selection based on Average Pearson Correlation Coefficient with Principal Component Analysis (APCC-PCA)

As mentioned in Section.I, in real-world seizure detection scenarios, non-seizure data is far more available than seizure data. A subject's first seizure might take a long time to occur, while ample non-seizure data is already collected. Consequently, online calibration should be performed using only non-seizure data before the first seizure happens, enhancing practical feasibility.

In the absence of calibration seizure data, the best alternative data may come from the source domain, in fact, because of the large number of individuals in the source domain, there is a high probability that there is a good alternative data. Because of the individual variability, the best way for select alternative data is to carry out online adaptive data selection through the calibration set non-seizure data that can be collected easily. Based on that, two main goals are proposed for looking for the best alternative seizure data. Firstly, the alternative seizure data should be very different from the non-seizure data of the calibration set, which can better make the difference between the two classes of templates large enough. Secondly, the non-seizure data of the individual selected should be little different from the non-seizure data of

the calibration set, because this can enable the model to have a high accuracy in the classification of non-seizure at the baseline before the MDAN training. The use of data in the two goals is also shown in Fig.3.

Based on the two goals, three types of features is selected for determining correlation of different data: 1. Time domain features included Mean Absolute Value, Mean Absolute Value Slope, Zero Crossings and Waveform Length. 2. Wavelet domain features which is the Wavelet Coefficient]. 3. Power spectral density (PSD). As for three types of features, the feature extraction is applied on every samples in the data. The above features are selected because many work have prove that during seizure and non-seizure, as well as between people, EEG features differ in the time domain, frequency domain, and wavelet domain. Therefore, the above characteristics can effectively characterize seizure and non-seizure EEG of different people[18][19][20].

---

**Alogrithm 1**  Calculation of APCC-PCA

**Input:** Seizure data of subjects in source domain $S_{iz}$ ; non-seizure data of subjects in source domain $S_{in-z}$ ; non-seizure data of calibration data $Tl_{n-z}$

**Output:** Average Pearson correlation coefficient of sliding window with PCA : $PCC_{a-pca}$

1:  Complete feature extraction of $S_{iz}, S_{in-z}, Tl_{iz}$

$FS_{iz}, FS_{in-z}, FTl_{n-z} = feature\_extraction(S_{iz}, S_{in-z}, Tl_{n-z})$

2:  Complete flatten and PCA of the extracted features

$FS'_{iz}, FS'_{in-z}, FTl'_{n-z} = PCA(Flatten(FS_{iz}, FS_{in-z}, FTl_{n-z}))$

3:  Segment samples in $FS'_{iz}, FS'_{in-z}$ with sliding window on time scale. The length of each segment of $FS'_{iz}, FS'_{in-z}$ equals to the length of $FTl'_{iz}$

4:  **For** Sengment $= 1,2,3,\cdots$, m of $F_{iz}, F_{in-z}$ **do**

5:  $\qquad PCC_1 = pearsonr(FS'_{iz,m}, FTl'_{n-z,m})$

6:  $\qquad PCC_2 = pearsonr(FS'_{in-z,m}, FTl'_{n-z,m})$

7:  **End For**

8:  $PCC_{a-pca} = mean(PCC_2 - PCC_1)$

---

As for each subject in source domain $S_i$ , its seizure data will be denoted as $S_{iz}$ ,while non-seizure data will be denoted as $S_{in-z}$ . And the non-seizure data of calibration data will be denoted as $Tl_{n-z}$ . As mentioned in Section III.A, $Tl_{n-z}$ is only 0.5% of the whole test set non-seizure data, which is much less than $S_{iz}$ and $S_{in-z}$ . After feature extraction, the feature sample number in $Tl_{n-z}$ is also much less $S_{iz}$ and $S_{in-z}$ . In order to calculate to overall feature correlation of the two different size data, average pearson correlation coefficient of sliding window with principal component analysis (APCC-PCA) is calculated, which is shown in Algorithm 1.

In Algorithm1, firstly, three types of feature extraction are first applied on every input data. Secondly, A PCA will be applied on every flattened features because of the high dimension. Because the dimensions of a single input window are too large (One input window contains 256 samples of 18 channels data, the detailed will be introduced in Section IV.A), the extracted features also have large dimensions. Specifically, for the time domain features, Wavelet Coefficient and PSD, 90, 2307, 4600 dimensions of features will be extracted for each input window respectively. Because the correlation will be calculated between every two window, the large dimensions would cost large computation, which may highly limit the time for online calibration after data collection. Therefore, PCA will be used to reduce the dimension to 64 for the three features. Thirdly, in order to calculate correlation of two different sample number of data, a segmentation with sliding window on time scale will be applied on features from source domain. At last, two Pearson Correlation Coefficient (PCC) will be calculated between every two window. $PCC_2$ is calculated for the goal that the alternative seizure data should be very different from the non-seizure data of the calibration set. And $PCC_1$ is calculated for the goal that the non-seizure data of the individual selected should be little different from the non-seizure data of the calibration set. $PCC_2-PCC_1$ represents the average performance of the two goals, and since the magnitude of the two PCC is close after average, no weight is used. The mean of all $PCC_2-PCC_1$ will be the last output. It should also be mentioned that different weight $W_i$ for gradient generated from different source domains is calculated from APCC-PCA, the APCC-PCA of the selected people will be normalized to be weight $W_i$ .

After calculation of (APCC-PCA), the top n subjects will be selected for both source domain training and template generating because of the relatively high correlation. (The effect of different n values on the final result will be shown in Section IV.B.)

## IV. EXPERIMENTS RESULTS

### A. Data Set Description and Experimental Setup

The Children's Hospital Boston-Massachusetts Institute of Technology (CHB-MIT) Scalp EEG Database [21] is applied in this paper. The CHB-MIT data set includes 686 EEG recordings from 24 individuals, with 198 seizures manually annotated by experts. All signals were sampled at 256Hz with a 16-bit resolution. The electrodes' positions and nomenclature adhered to the international 10-20 system. The EEG recordings contained channel number of 18 or 23. For the 23 channel recordings, the 18 channels are also included and the same.

Based on that, we extracted all 186 seizure recordings that included the 18 channels mentioned above, amounting to approximately 3 hours of data, along with around 44 hours of corresponding non-seizure recordings. For the training dataset, the EEG signals were divided into1-second segments without overlapping. For the test data set, a sliding time window of 1 second with a overlap size of 0.875 seconds is applied. Majority vote is also applied with the size of 33.With majority

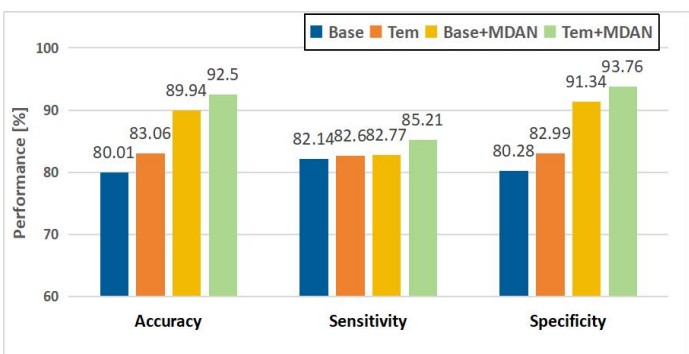

Fig. 4. Performance of accuracy, sensitivity and specificity with proposed template matching and MDAN.

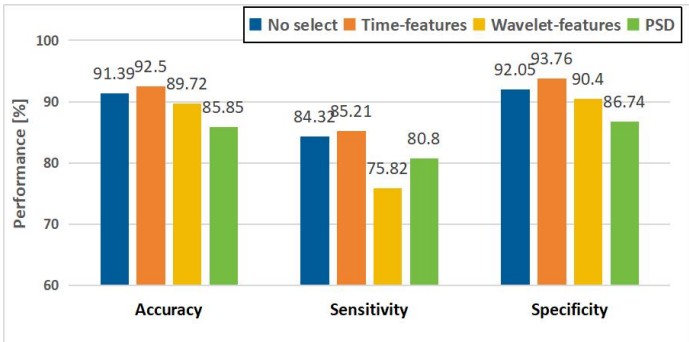

Fig. 5. Performance of accuracy, sensitivity and specificity with subjects selected by the proposed APCC-PCA.

vote, the classifier independently analyzes the signal and casts a vote for a particular class. The class that receives the majority of votes is chosen as the final classification. Majority vote has a good effect on seizure detection with a certain duration of onset and non-onset

The commonly used leave-one-out cross-validation is also applied for cross-individual seizure detection[10]. With total 24 individuals, data from 23 individuals is used for train set, while data from the remaining individual is used for test set. The overall sensitivity, specificity, and accuracy are calculated by averaging the results from all 24 test scenarios.

The training hyper-parameters of neural network are set as below. Sparse categorical cross-entropy is used for both label classifier and domain classifier as loss function. The learning rate is set to 1e-3 with 1e-6 decay, epoch for both baseline model and MDAN are set to 20 and batch-size for both baseline model and MDAN are set to 48.

*B. Results*

In order to show the improvement brought by template matching and MDAN, Fig.4 shows the accuracy result of baseline model and template matching. "Base" denotes accuracy of baseline model which the classification is completed directly using the neural network, and the result is obtained through the probability calculation of the output of last fully-connected layer. "Tem" denotes accuracy of using template matching for classification. "Base+MDAN" denotes

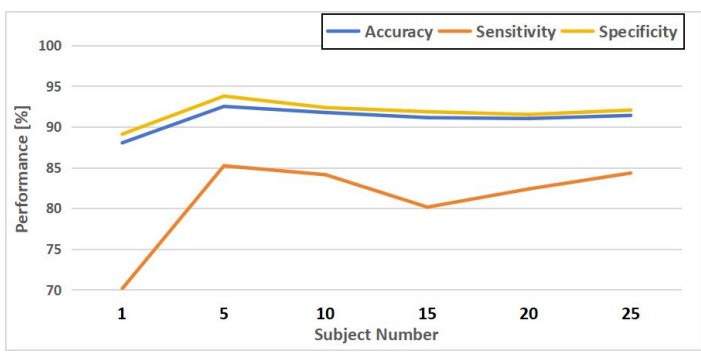

Fig. 6. Performance of accuracy, sensitivity and specificity with different number of subject selected by the proposed APCC-PCA.

accuracy of baseline model after MDAN training. "Tem+MDAN" denotes accuracy of template matching after MDAN training. All the results here used adaptive data selection with the best features which will be introduced in the next paragraph. The results indicate that the "Tem" method is 3.05%, 1.46% and 1.71% higher than that of the "Base" method on accuracy, sensitivity and specificity, and the "Tem+MDAN" method is 2.56%, 2.44% and 2.42% higher than that of the "Base+MDAN" method on accuracy, sensitivity and specificity. It can be seen that template matching can improve the accuracy both with or without MDAN. And based on template matching, with MDAN achieves 8.44%, 2.61% and 8.77% of improvement on accuracy, sensitivity and specificity with no MDAN. The result shows that MDAN can effectively improve both three performance.

In order to show the improvement brought by adaptive data select for non-seizure supervised transfer learning, Fig.5 shows the accuracy of using source domain subjects chosen by different types of features. "No select" denotes accuracy of model using all the source domain subjects for training and as alternative seizure calibration data. "Time-features" denotes accuracy of using top 5 subjects chosen by time-features. "Wavelet-features" denotes accuracy of using top 5 subjects chosen by wavelet domain features. "PSD" denotes accuracy of using top 5 subjects chosen by power spectral density. The results shows that with time-features selected, it reaches the best accuracy, sensitivity and specificity compared with no select and other features. Therefore time domain features is chosen to be the best features and applied in final result.

However, the number of subjects chosen can make a big difference in accuracy. Fig.6 shows the accuracy of using different number of source domain subjects selected by time features. It should be mentioned that the number of domain classifier equals to the subject numbers here. The results shows that with 5 subject number, the result reaches the best. Therefore, 5 is chosen as subject number in final result.

The benchmark table is shown in Table I. From Table 1, it can be seen that the result of our work outperform other work in the three performance metrics especially accuracy and sensitivity.

TABLE I.    BENCHMARK TABLE

| Work | Method | Sensitivity(%) | Specificity(%) | Accuracy (%) |
|------|--------|------------|------------|-----------|
| [13] | CNN, RNN | 85.00 | - | - |
| [12] | KNN | - | - | 74.03 |
| [9] | KNN | 88.00 | 88.00 | - |
| [16] | SOF | 84.67 | 82.06 | - |
| [10] | CNN+TM | 84.75 | 92.31 | 91.92 |
| **Our work** | **MDAN+TM+** | **85.21** | **93.76** | **92.50** |

## V. CONCLUSION

In this work, two methods are proposed. Firstly, by treating cross-individual tasks as transfer learning scenarios, template matching based on multiple source domain adaption neural network method is proposed. Secondly, considering the fact that seizure data is much less than non-seizure data, adaptive calibration data select based on average Pearson correlation coefficient with principal component analysis method is proposed. On CHB-MIT data set, method proposed achieves a state-of-the-art performance of 85.21% sensitivity, 93.76% specificity and 92.50% accuracy.

## ACKNOWLEDGMENT

Grateful for the support from the National Key R&D Program of China under Grant No. 2021YFB3200601.

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
