# OpenReview forum: "A High Cross-Individual Accuracy EEG-based Seizure Detection Algorithm Based on Multiple Source Domain Adaption"
_IEEE.org/ICIST/2024/Conference — IEEE ICIST 2024 Conference Submission_

### Official Review · Reviewer_TrW4 · 2024-09-01
**Good paper, accept**

**Rating:** 7
**Confidence:** 4

**Review:**

1. Please explain the significant benefits of using EEG to detect whether a patient with epilepsy is experiencing a seizure. People around the patient can determine whether the patient is experiencing a seizure based on his condition, and providing urgent care for him having a seizure may be a more critical matter.
2. The authors need to consider the time complexity of the proposed methods and provide a comparison with other methods.
3. The presentation quality of this paper requires significant improvement. The authors are advised to carefully review and correct several typographical and grammatical errors within this paper.
4.  The authors should explicitly outline the motivation and contributions of this paper in comparison to existing works in the field.
5. Please clarify the superiority of the proposed method.

---

### Official Review · Reviewer_WmtM · 2024-09-01
**Comments to paper 53**

**Rating:** 9
**Confidence:** 4

**Review:**

Due to differences in brain structure and skin, EEG signals can vary greatly from one individual to another, which leads to a low cross-individual seizure detection accuracy. To solve the problem, two methods are proposed in this work.  With the method, the accuracy further improved. Some comments should be considered.
1. Which is the training data, which is the validation data?
2. Which is the architecture of the neural network?

---

### Decision · Program_Chairs · 2024-09-06

Accept (Oral)